# Deriving atmospheric turbulence intensity from profiling pulsed lidar measurements

Maxime Thiébaut<sup>1</sup>, Marie Cathelain<sup>1</sup>, Salma Yahiaoui<sup>2</sup>, and Ahmed Esmail<sup>3</sup>

<sup>1</sup>France Énergies Marines, Technopôle Brest-Iroise, 525 Avenue Alexis de Rochon, 29280 Plouzané, France
 <sup>2</sup>Vaisala France SAS, 6A, rue René Razel, Tech Park, CS 70001, 91400 Saclay Cedex, France
 <sup>3</sup>GL Garrad Hassan Deutschland GmbH, Loads & Power Performance & Wind Resource (E-NX-RL), Sommerdeich 14b, 25709 Kaiser-Wilhelm-Koog, Germany

Correspondence: Maxime Thiébaut (maxime.thiebaut@france-energies-marines.org)

Abstract. A new method is proposed to provide estimates of the turbulence intensity (TI) from measurements of pulsed lidars (light detection and ranging) employing the Doppler beam swinging technique. This method relies on combining the variances of the line-of-sight (LOS) velocities collected by the five independent beams of the lidars and, as such, is referred to as the variance method. The variance method comes with an explicit removal of the Doppler noise (inherent to the instrument) to

- 5 the variance of the LOS velocities. Turbulence metrics derived from the variance method are compared to that derived from a standard method, commonly used in the wind energy industry. Reference turbulence measurements are provided by a sonic anemometer mounted on a meteorological mast, installed nearby the lidars. Two configurations of the WindCube v2.1 lidars are proposed: the commercial configuration and a prototype configuration, sampling 4 times faster, thus allowing to capture the turbulent energy of smaller eddies. The standard method applied on wind measurements collected by both configurations
- 10 shows mean errors in TI estimates of more than 50%. The application of the variance method on measurements collected by the commercial and prototype configuration drops the mean error to 16.7% and 13.2% respectively.

#### 1 Introduction

A comprehensive assessment of the inflow conditions is highly important for an optimal planning and design of wind energy projects. More specifically, the measurement of the ambient turbulence is critical to gain confidence in the representativeness

- 15 of the aerodynamic loading on wind turbine structures and the modeling of future wind farms effects, i.e., impacts on the atmospheric flow by the single wind turbines, within wind farms, and large clusters (Rohrig et al., 2019; Veers et al., 2019). In particular, the knowledge of the turbulence intensity, TI, is essential since TI is directly involved in modeling applications of wake effects within wind farms which can significantly decrease the power production and increase the cost of electricity (Howland et al., 2019). Turbulence is also known to affect the lifetime of certain turbine components (e.g., blades, rotor, tower)
- 20 and thus, are highly relevant for questions of potential lifetime extensions (Kelley et al., 2005, 2006). Moreover, wind turbine performance is impacted by turbulence levels and wind shear driven by atmospheric stability (Wharton and Lundquist, 2012; Clifton and Wagner, 2014).


In the wind power industry, the characterization of ambient turbulence through measurements of meteorological mast anemometry is the traditional method. Either cup or sonic anemometers are mounted on slender booms at several altitudes over a certain period of time. However, wind turbines have experienced a continuous growth in size in the past decades and the upper tip of onshore wind turbines blade can now easily reach heights up to 200 m above the ground. This increases the needs for wind and turbulence measurements at higher altitudes. Collecting measurements at such height with meteorological masts is non longer a viable solution since installing and operating such massive infrastructures is cost-prohibitive.

In response, remote sensing devices such as profiling lidars (light detection and ranging) have recently emerged as alterantives to meteorological masts. Measurement methods used by lidars are fundamentally different than those used by cup or sonic anemometers. Anemometers provide an estimate of the wind speed across a volume a few cubic centimeters whereas lidars provide an average across a probe volume of up to several dozen cubic meters. This system can be categorized according to their emission waveform, i.e., pulsed or continuous, and measuring technique, i.e., Doppler beam swinging (DBS) (Strauch et al., 1984) or velocity-azimuth display (VAD) (Browning and Wexler, 1968). Lidar systems offer the potential for reduced costs compared to meteorological masts and the ability to measure at the same or even greater heights above the ground.

While profiling lidars have proven to be accurate tools for measuring mean wind speed and direction (Smith et al., 2006; Emeis et al., 2007; Sjöholm et al., 2008; Wagner et al., 2011; Gottschall et al., 2012; Kim et al., 2016) they are still not accepted for turbulence measurements which is an active area of research (Sathe et al., 2015; Newman et al., 2016). This lack of acceptance is mainly due to, (1), large measurement volumes leading to spatial averaging of turbulence along the line-

of-sight (LOS), (2), cross-contamination by different turbulent structures of the wind field, (3), low sampling rate, and , (4), instrument noise. The present paper will address the second, third, and fourth limitations.

The spectral signature of the cross-contamination effect has been clearly identified in turbulent kinetic energy (TKE) spectra, i.e., velocity variance as a function of frequency, by an almost complete attenuation of the turbulence signal in the inertial subrange (Canadillas et al., 2010; Sathe and Mann, 2012; Kelberlau and Mann, 2020) preceded by a hump of energy generated by

the beam interference phenomenon. This phenomenon induces additional variance to the signal which causes an overestimation of TI.

In this paper, the low sampling rate is tackled by proposing, for the first time, the deployment of a WindCube v2.1 lidar with a sampling rate four times higher than the commercial technology. This faster sampling rate configuration is expected to capture smaller eddies and their associated turbulent energy thus allowing for more accurate estimation of turbulence. Synchronous

measurements of the lidar operating with a faster sampling rate configuration, alongside with measurements of a commercial lidar is proposed to assess the benefit of the new configuration.

Finally, the instrument noise correction is addressed. Among the four limitations of the lidars, instrument noise correction is less explored and, as far as we know, not documented in the literature. This specific topic could benefit from recent works carried out in the field of ocean engineering. In ocean science, acoustic Doppler current profilers (ADCP) are often the standard

instrument for measuring flow speed and turbulence at different depth levels, throughout the water column. Recently, five-beam ADCP has been commercialized (Guerra and Thomson, 2017). The configuration of such sensors is similar to the WindCube v2.1 lidars configuration with four-diverging beams and a fifth beam pointing vertically upward. Both five-beam ADCPs and

Figure 1. Left: top view of a WindCube v2.1 lidar with positions of its five beams. The x component is oriented from beam 3 towards beam 1, the y component points from beam 4 towards beam 2, and the vertical z component points downwards along Beam 5. Right : tridimensional view with the angle of beam inclination  $\theta = 28^{\circ}$ .

WindCube v2.1 lidars employ the Doppler effect through backscattered signal of emitted pulses in order to measure the flow velocity at different heights above instrument. However, measurements based on Doppler effect are affected by the Doppler noise, inherent to the instrument, which produces significant overestimation in the calculation of turbulence metrics. In the 60 field of ocean engineering, this effect has been clearly identified and methods have been developed to correct turbulence measurements from the Doppler noise-induced variance resulting in a substantial improvement of the turbulence estimation (Thomson et al., 2010, 2012; Richard et al., 2013; Durgesh et al., 2014; Thiébaut et al., 2020a, b). Considering similarities in the measurements principles between ADCPs and pulsed lidars, it is expected that Doppler noise also strongly affects turbulence estimates derived from measurements of pulsed lidars.


In this paper, a method used for correcting Doppler noise of oceanographic measurements is transposed to improve atmospheric measurements. The method is tested on simultaneous measurements collected by two pulsed lidars employing the DBS technique with two different sampling rates. Reference turbulence measurements are provided by a sonic anemometer mounted on a meteorological mast, installed nearby the lidars. The focus is on the altitude matching the position of the sonic

anemometer. Variances of the LOS velocities provided by the lidars are corrected from the Doppler noise-induced variance and combined through trigonometric expressions proposed by Dewey and Stringer (2007), originally dedicated to ocean engineering applications. This method relies on combining the variances of velocity measurements collected by five independent beams and, as such, is referred to as the variance method. This method attenuates the cross-contamination effect and allows for a considerable improvement of turbulence metrics.

#### Data collection and methods 75 2

#### 2.1 WindCube v2.1 - commercial and prototype configuration

The measurement of wind speed and direction derived from a commercial WindCube v2.1 lidar is based on the pulsed Doppler heterodyne laser principle. The lidar sends tenths of thousands 175 ns long laser pulses in the atmosphere. Light pulses are backscattered by aerosols and, from those received, Doppler shift is analyzed before the next laser pulse is emitted. This avoids confusing time delays and distances. Therefore, probe distance, or height, only depends on the time it takes for a pulse to be

received after it has been emitted.



The WindCube v2.1 enables wind profile measurement from 40 meters to 300 meters through up to 20 independently configurable measurement heights. The WindCube v2.1 technology ensures a 20-m long constant probe volume of atmosphere which leads to constant accuracy at all heights. Four beams are sent successively in four cardinal directions along a 28° scanning

85 cone angle, followed by a fifth, pointing vertically upward and providing a direct estimate of the vertical velocity. The LOS velocities (i.e, radial velocities) are measured simultaneously at each configured height before switching to another beam. The sampling rate of the LOS velocities of the commercial WindCube v2.1, hereinafter referred to as "commercial configuration", is  $f_c = 0.25$  Hz. The sampling rate,  $f_p$ , of the LOS velocities of the new configuration, hereinafter referred to as "prototype" configuration", is four times faster, i.e.,  $f_p = 1$  Hz.

#### Coordinate system and preliminaries 90 2.2

The LOS velocities are combined to calculate the horizontal components ( $V_x$  and  $V_y$ ) of the wind speed in the instrument coordinate system which is a left-handed coordinate system defined by the beam directions. The x component is oriented from beam 3 towards beam 1, the y component points from beam 4 towards beam 2, and the vertical z component points downwards along Beam 5 (Fig. 1). Defining the wind field in the direction of beam i as  $b_i$ , with positive velocity being towards the instrument, the coordinate transformation from beam coordinates to instrument coordinates is given by Eq. 1 and 2:

$$V_x = \frac{b_1 - b_3}{2\sin\theta} \tag{1}$$

$$V_y = \frac{b_2 - b_4}{2\sin\theta} \tag{2}$$

where  $\theta = 28^{\circ}$  is the angle of divergence of each beam position from the vertical, i.e., the beam inclination angle.

#### 2.3 Turbulence intensity 100

The turbulence intensity, TI, is the most common metric used in the wind energy industry as well as other engineering fields in order to quantify turbulence. TI is referred to as the turbulence level and represents the intensity of velocity fluctuations. TI