# Peer review of "Deriving atmospheric turbulence intensity from profiling pulsed lidar measurements"

_Wind Energy Science, 2022_

## Referee Comment (RC1)

**Deriving atmospheric turbulence intensity from profiling pulsed lidar measurements**

Reviewer's comments

**1 General comment**

The manuscript "Deriving atmospheric turbulence intensity from profiling pulsed lidar measurements" by Thiébault et al. deals with the estimation of the turbulence intensity of the horizontal wind velocity component using the 5-beam DBS scanning mode of the WindCube v2.1 lidar. The manuscript focuses first on the turbulent kinetic energy (TKE) spectrum and then assesses the accuracy of the different methods and lidar configurations to study the turbulence intensity using various metrics.

The manuscript addresses challenging, contemporary and interesting questions about turbulence measurements at heights relevant to large wind turbines. Therefore, the paper is of broad international interest and within the scope of Wind Energy Science (WES), although it may be better suited to Atmospheric Measurement Techniques (AMT). The paper brings a novel idea for the calculation of the velocity variance, inspired by ocean science, which is welcome. The objective and methods are clearly outlined.

However, the analysis and method need some improvements as they sometimes rely on unverified or incorrect assumptions. Some preliminary steps lack rigour. More critically, the discussion and results are sometimes self-contradictory, for example, the discussion on the probe length. The content can be more concise. For example, section 3.1 (turbulent kinetic energy spectra) diverges sometimes from the initial scope of the paper. Section 3.3 could be summarized in a couple of sentences. Even if the paper is smoothly written and read quite easily, a major revision is, therefore, needed. In summary, the manuscript relies on a good idea, but the method, results and discussion need considerable improvements to be considered for publication in WES.

**2 Specific comments**

**Point 1**

The turbulence intensity (TI) can be calculated for the three wind velocity components: $I_u$, $I_v$ and $I_w$ with $I_u > I_v > I_w$. For wind turbine design, the along-wind component $u$ and cross-wind components $v$ are the most important. In the manuscript, eq (3) gives the turbulence intensity of the horizontal components, i.e. $u^2 + v^2$, which is of limited interest for wind turbine design. Therefore, I suggest focusing on the $u$ and $v$ components separately.

**Point 2**

The manuscript correctly mentions that the TI is a central and practical metric for the study of atmospheric turbulence. However, the velocity spectrum is ultimately needed for the wind loading calculation. Therefore, the importance of the TI for wind energy and wind engineering should not be overestimated. This aspect may be pointed out in the introduction of the manuscript.

**Point 3**

Line 29-30: I suggest not to write that lidars have emerged as an alternative to met masts but rather as a "complementary tool". This is particularly true for turbulence measurements: both profilers and scanning Doppler wind lidars are currently unable to match the performances and data availability of 3D sonic anemometers.

**Point 4**

Line 41: The manuscript argues that the along-beam spatial averaging effect will not be addressed. I think it should, on the contrary, be studied. Several of the questions raised by the manuscript can be linked to the fact that the along-beam spatial averaging is not discussed jointly with the sampling frequency, for example.

**Point 5**

In the manuscript, arguments for using a higher sampling frequency are given. I agree that increasing the sampling frequency $f_s$ should helps reducing the statistical uncertainties (see e.g. Kaimal and Finnigan (Section 7.1, 1994)). However, because of the along-beam spatial averaging of 20 m or more, increasing $f_s$ to values higher than 1 Hz (or even 0.25 Hz?) may not help to capture smaller eddies.

**Point 6**

Section 2.2: A sketch to show the different wind velocity components in the wind-based coordinate system ($u$, $v$ and $w$) and the lidar-based coordinate system $V_x$, $V_y$, $V_z$ would be a good idea for the sake of pedagogy.

**Point 7**

Section 2.3.2: It is a little unclear to me where the pitch and roll angle comes from. I understand that these angles are based on the publication by Dewey and Stringer (2007) about the ADCP system, but I don't know if these are necessary for a Doppler wind lidar. If yes, how were they measured? were they non-negligible?

**Point 8**

As far as I know, the study by Dewey and Stringer (2007) is actually unpublished. The reference used in the manuscript seems to be given by Google Scholar, which may be incorrect.

**Point 9**

Eq. 10 seems to be unnecessary complex since one could argue that $w \approx b_5$. Section 3.3. of the manuscript seems to show that assuming $w \approx b_5$ is good enough. Therefore, I suggest moving and squeezing the content of section 3.3. right after equation 10. In general, one wants to avoid presenting an excessively heavy formula for the sake of clarity.

**Point 10**

Section 2.4.1: The company Thies Clima produces different types of sonic anemometers. It is unclear to me which model was used. Note that if a 2D sonic anemometer was used, one should keep in mind that this type of sensor is mainly used as a weather station. So this sensor is not expected to perform as well as a 3D sonic anemometer to study turbulence. This is an important point to remember if the goal is to validate Doppler wind lidar turbulence measurements against sonic anemometry.

**Point 11**

Fig 2 contains elements that are too small to be easily readable. I suggest redrawing it. Ideally, the figure should indicate whether the sonic anemometer is on the northwest or southeast side of the tower. From the text, it seems it is on the southeast side. Maybe a digital Elevation Map can be used instead of satellite images? Otherwise, it can be difficult to visualize the local topography.

**Point 12**

Line 170: The comparison between sonic anemometer data and the lidar data is done using the same sampling frequency $f_s$ (0.25 Hz and 1 Hz). I agree with the authors on this approach. However, it also means that the turbulence intensity will be underestimated compared to a standard 3D sonic anemometer using $f_s > 10$ Hz. This should be clarified in the manuscript.

**Point 13**

Line 172-178: There is no guarantee that a 10 min duration is short enough to ensure stationarity of the velocity fluctuations. To assess the assumption of stationarity, a trend test (non-parametric) or a parametric test should be used.

**Point 14**

Line 172-178: The standard deviation $\sigma_H$ of the horizontal wind velocity component $H$ is studied using an averaging time of 10 min. For turbulence measurements, averaging times of 30 min to 60 min are often used. Considering the Simiu and Scanlan spectrum for the along-wind component $u$, it can be shown that using 10 min instead of 60 min (30 min) will lead to an underestimation of $\sigma_u$ by 15% (11%).

The standards and codes for wind turbine design use often 10 min (i.e. 600 s) as averaging time because the wind loading is computed utilizing the velocity spectra. Considering frequencies down to $\frac{1}{600\,\text{s}} = 0.00167$ Hz is often good enough to describe the full range of vibrations of a large engineering structure. However, it does not mean that 10 min is long enough to study integral turbulence characteristics.

**Point 15**

Line 186: Does the 75% data availability means that time series with 25% or less of NaNs were kept? if yes, this may be too much. I suggest dismissing samples if the percentage of Nans is 10% or higher. In atmospheric science, the acceptable percentage of NaNs is usually from 2.5% to 5%, but this may be too strict for Doppler wind lidar data.

**Point 16**

Line 187-188: I do not understand the sentence "A percentage ranging [...] was rejected". May it be possible to clarify it?

**Point 17**

Section 2.6: For the study of the gradient Richardson number, I recommend using the potential temperature $\theta$ or virtual potential temperature $\theta_v$ instead of the absolute temperature $T$. From figure 2 in the manuscript, the pressure and humidity seem to be measured at 10 m and 95 m, so $\theta$ and $\theta_v$ may be calculated successfully. It may also be a good idea to use a more accurate classification of atmospheric turbulence than $R_i < 0$ or $R_i > 0$, which could be too rough.

**Point 18**

Section 3.1: This section contains some unnecessary sentences (lines 211-215) and may have to be re-written to better anchor it to the research question. Figure 3 should be redrawn. It is vital to better highlight the presence (or absence) of an inertial subrange. Because of the low-sampling frequency, the cross-contamination and the spatial averaging, the inertial subrange may not be easily visible.

To improve the visualization of the velocity spectra: (1) the velocity spectra should be split between $u$ and $v$ components, and (2) the velocity spectra should be pre-multiplied with the frequency or wavenumber. If needed, you can further normalize the spectra using the variance $\sigma_u^2$ measured by the sonic anemometer at 95 m; (3) The frequency should be replaced by the

wavenumber or a normalized frequency (4) the velocity spectra should be split between unstable or stable stratifications.

**Point 19**

Line 236-243: These lines seem unnecessary. They could be removed without affecting the content of the paper.

**Point 20**

Have you tried estimating the standard deviation of the $u$ and $v$ components by integrating a fitted empirical velocity spectrum as

$$\sigma_u^2 = \int_0^\infty S_u(f) d_f \tag{1}$$

where $f$ is the frequency in Hz and $S_u$ has the following form:

$$f S_u(f) = \frac{A f_r}{(1 + B f_r)^{5/3}} \tag{2}$$

$$f_r = \frac{f z}{\overline{u}} \tag{3}$$

where $A$ and $B$ are empirically obtained by least-square fit to the estimated spectrum and $\overline{u}$ is the mean wind speed at height $z$. If yes, how does this method compares with the noise-removal approach adopted in the present study? Note that you may have to apply a method similar to Kelberlau and Mann (2020) to reduce the cross-contamination in the velocity spectrum.

   Following Kelberlau and Mann (2020), the velocity spectra from DBS scans should not be fitted by turbulence models. Although they do not specify what they call "turbulence models", one can assume they refer to the uniform shear model or 3D isotropic spectral turbulence models. I ignore if their recommendation for "turbulence models" includes eq. (2). Nevertheless, attempting to estimate $\sigma_u$ using eq. (1) and eq. (2) may be worthwhile, especially when the lidar system is aligned with the mean wind direction.

**Point 21**

Fig 4: It is unclear to me what the figure aims to demonstrate. If the quadratic relationship is not used elsewhere in the manuscript, should it be kept? Is the $R^2$ value shown in this figure the Pearson or Spearman correlation coefficient? I suggest using the Spearman correlation coefficient if the relationship between the two variables is non-linear monotonic. Alternatively, the RMSE can be used instead.

**Point 22**

Fig 6 does not seem to clearly support the conclusions of the manuscript. Was it because there was no noise removal here? I encourage the use of a colormap that is perceptually uniform instead of the jet colormap.

**Point 23**

Section 4: The discussion seems to recommend a large probe length to study turbulence. In general, when turbulence is studied with a Doppler wind lidar, one wants the probe volume length to be as small as possible. A 20 m probe volume length is already quite large, which is the reason continuous-wave Doppler (scanning) lidars with a probe volume length smaller than 10 m have a higher potential than pulsed lidar to study turbulence. Therefore, this section may need some reformulations.

**Point 24**

Lines 381-394: This paragraph seems more adapted to the beginning of the manuscript since it reviews some previous results. Maybe this can be moved there and shortened?

**Point 25**

The conclusion may have to be reformulated as it includes several recommendations that could be criticized. I agree that operating a lidar at a higher sampling rate is a good idea, but this will not be useful if the probe volume is not reduced. As pointed out by the authors, reducing the probe volume increases, in return, the measurement noise. So the situation is rather complex. Does it mean that DBS scans should only be used to study the mean flow characteristics? Should new scanning modes be developed instead?

Also, the conclusion mentions the use of lower beam inclination $\theta$ to improve the measurement accuracy. This may be a good idea. However, if $\theta$ becomes too small, the measurement uncertainty will increase substantially because the angle between the beam and the horizontal direction will get close to $90°$. So the ideal value of the beam inclination theta is not trivial either. Do you have any specific value in mind?

**Point 26**

The manuscript contains some typographical errors. I recommend a quick proofread. The online web app Grammarly (`https://app.grammarly.com`) is quite good for this purpose.

**References**

Kaimal, J.C., Finnigan, J.J.. Atmospheric boundary layer flows: their structure and measurement. Oxford university press; 1994.

Dewey, R., Stringer, S.. Reynolds stresses and turbulent kinetic energy estimates from various adcp beam configurations: Theory. Unpublished 2007;.

Kelberlau, F., Mann, J.. Cross-contamination effect on turbulence spectra from doppler beam swinging wind lidar. Wind Energy Science 2020;5(2):519–541.

---

## Referee Comment (RC2)

**Review for manuscript wes-2022-53**

**August 9, 2022**

Thiebaut et al. describe a new method to estimate turbulence intensity from profiling lidar measurements. Such studies are quite relevant for site assessment in wind energy. Better estimates of turbulence intensity are very important for load predictions. It is well known that profiling lidars have some shortcomings when it comes to turbulence estimation and new methods to improve the results are thus welcome. The idea in this manuscript to use methods from acoustic Doppler current profilers (ADCP) which are used in oceanic sciences is interesting and attractive. Three main changes are made to commercial profiling lidars: the sampling rate is increased, the variance method as used with ADCPs is implemented and a noise removal is applied. While the increase of sampling rate is straight forward and the explanations are easy to follow, the critical differences between ADCP and profiling lidars for atmospheric measurements are not sufficiently well elaborated. For this reason, I cannot suggest the paper for submission in WES before some major revisions.

**1 General comments**

1. The main difference between ADCP and lidar DBS is the character of the atmosphere, which can be much more instationary than ocean currents, and the fact that a lidar DBS does not measure all beams simultaneously. A minor difference is that the onshore lidar as it is used in this study does not move, so it has a constant roll and pitch angle which equals zero in best case. It is unclear why the authors use the full roll and pitch equation from Dewey and Stringer (2007) and not the much more simple zero pitch and roll equations. From those equations, it would also be very easy to check if the assumptions of homogeneity and stationarity are valid for the dataset by checking the equations 11&12 from Dewey and Stringer against the sonic anemometer data:

$$\overline{b_1'^2} + \overline{b_2'^2} \quad = 2\overline{u'^2}\sin^2\theta + 2\overline{w'^2}\cos^2\theta \tag{1}$$

$$\overline{b_3'^2} + \overline{b_4'^2} \quad = 2\overline{v'^2}\sin^2\theta + 2\overline{w'^2}\cos^2\theta \quad . \tag{2}$$

Does the left hand side as measured from the lidar DBS equal the right hand side as measured from the sonic anemometer for different wind speeds

and different stability conditions? Only checking the vertical wind is not sufficient.

2. The definition of Doppler noise seems to be a quite vague to me. Parts of the explanation are given in different parts of the manuscript, but it should be better introduced in the beginning. It is not true that lidar noise has not been studied and included in models for turbulence retrieval from lidars before. The work by R.G. Frehlich as well as I. Smalikho and E.J. O'Connor contain much information about lidar noise. The noise which is meant here should be put into context to these other studies.

3. The way that stability is derived from the Richardson number here is not correct for atmospheric sciences. What is calculated is the bulk Richardson number, which has a non-zero critical value between stable and unstable flow. This has an impact on the whole comparison between stable and unstable conditions in this study and needs to be revised.

**2 Specific comments**

- p.2,l.40: what about availability?

- p.2, ll.51f: lidar noise is explored quite extensively in works by R.G. Frehlich, I. Smalikho and E.J. O'Connor. These works should be reflected.

- p.4,l.83: "volume of atmosphere"? This is a strange expression.

- Fig.2: basic information of the maps are missing: scale, northing, lat/lon, copyright of the map pictures.

- p.8,l.170: I do not understand the reason for the resampling of the sonic. Is it not the goal to compare the lidar derived TI to the best possible measurement? If the sonic is downsampled, it will also lack very small scale turbulence. Probably the difference is not much, but I would suggest to use the best possible TI estimate for comparison, unless you want to isolate the errors by specific processing steps.

- p.8,l.174: it is a questionable statement that 10-minutes are enough to retain the longest ime scales of coherent turbulent structures. In basic ABL research, 30-minutes are rather the standard.

- p.8, l.197: It is too much simplified to use the measured temperature difference and standard atmosphere lapse rate for the calculation of the Richardson number. Please calculate the potential temperature difference.

- p.9, l.205: It might be partly because of the way the Richardson number was calculated, but it is certainly a strong overestimation if 89.4% of the data are considered to be measured in a stable atmosphere. What is actually calculated here is the bulk Richardson number which has a critical value of 0.25 for unstable flows.

- p.11, ll.242f: I am less concerned about the inertial subrange being present at frequencies higher than the Nyquist frequency, but more concerned if the integral length scale is large enough to yield an inertial subrange at the resolved frequencies, especially in stable atmospheric conditions. Typically, for lidar turbulence retrievals, there is a minimum integral length scale below which the estimates need to be discarded.

- p.11, l.247: does $N_i$ and $n_i$ have to be determined for each beam individually? How much do they differ and why?

- p.11, l.248: this seems to be a random choice for the characteristic frequency. Can this be justified by error quantification?

- p.11, l.252: I assume you mean Eq. 5.

- p.17, l.363: what does "a frequency domain 9 times wider" mean?

- p.18, ll.370ff: These explanations and theories would be much easier to follow and understand if the Doppler noise was presented in a concise mathematical formulation before.

- p.19, l. 411: Reducing the beam spread needs to be carefully traded off against horizontal wind speed retrieval accuracy.

---

## Referee Comment (RC3)

**Review of "Deriving atmospheric TI from profiling pulsed lidar measurements"**

Thanks a lot for your manuscript. I have several mayor comments as well as minor comments that I think are mostly related to misconceptions with regards to the nature of measuring turbulence with lidars and lack of clarity.

**Major comments**

1. In the abstract, as an example, the authors say that a new method is proposed to estimate TI from pulsed profilers. First, the method the authors used was proposed by Dewey and Stringer [2007]. Second, the use of beam variances have also been used before to estimate velocity-component variances at the least by both profilers and nacelle-based lidars [Sathe et al., 2014, Fu et al., 2022]. So the novelty should be clarified.

2. It could be the writing style but I am not sure whether the authors understand the problem of measuring turbulence with a lidar. For example, the cross-contamination is not due to the different structures of the field (line 40) but due to the influence of different velocity components on the line-of-sight variance, which is a result of the lidars scanning strategy. Also, in line 45 they say that cross-contamination causes and overestimation of TI. That is not always true; it might happen but that depends on how much filtering due to probe volume averaging you have. So, in many cases, if not most cases, filtering is the biggest threat in lidars.

3. Also important is that by acquiring the velocity faster the change in the variance should not be high and so neither in the TI. You will increase the uncertainty on the variance by measuring slower but the bias should not change unless you definitively and systematically are missing fluctuations by the turbulence structures but this is not the case of this unit measuring this non-complex flow.

4. Instrument noise corrections have been explored before (line 53) [e.g., Peña et al., 2017]. Perhaps you could explore that method as in that work they study a pulsed lidar too. This would avoid using a threshold to establish the frequency at which you expect noise, which is what I think you are doing. Your method does seem very sensitive to this choice and you should show how sensitive or not indeed is.

5. You have defined TI to be the parameter you want to look for. However, you should also present mean and variance comparisons of the velocity components as the problems with turbulence should be clearer seen when computing the variance and you want to make sure both lidars measure the same mean wind as your sonic. Also in wind energy the TI is normally defined based on the along-wind component or horizontal wind. If you have your fixed lidar beams then the horizontal velocity variance $\sigma_S^2$ is not $\left(\sigma_x^2 + \sigma_y^2\right)/2$ as you imply but $\left(V_x^2\sigma_x^2 + V_y^2\sigma_y^2\right)/S^2$ where $S$ is the horizontal velocity magnitude in the case the covariance between the two horizontal components is assumed zero (which is probably ok in your case). So I wonder why you choose to define TI like this. Perhaps you can make a comparison of sonic variance as you imply (half of the sum of both variances) against deriving the variance from the horizontal velocity time series from the sonic.

6. My most important comment is the use of the method by Dewey and Stringer [2007]. I cannot find what values of pitch and roll the authors use in Eqs. (8)–(10). If they use close to zero values (as I guess they place the lidar relatively well), then those equations are not needed but the classical Eberhard et al. [1989] expressions for the beam velocity variance, which one can easily show lead to:

$$\langle u'u' \rangle = \left(\sigma_3^2 + \sigma_1^2 - 2\sigma_5 \cos^2\theta\right) / \left(2\sin^2\theta\right), \tag{1}$$

$$\langle v'v' \rangle = \left(\sigma_2^2 + \sigma_4^2 - 2\sigma_5 \cos^2\theta\right) / \left(2\sin^2\theta\right) \tag{2}$$

For $\langle w'w' \rangle$ you do not need an expression as you measure with a vertical beam (so you can see that when $\phi_1$ and $\phi_2$ are close to zero in Eq. (10), the variance of $w$ and that of beam 5 are the same).

7. Section 3.1. makes little sense to me. First in Fig. 3 you seem to be plotting the beam power spectra and not the TKE spectra as you state in the text. Second, and most important, you are here analyzing the slope of the spectra within the frequency range where the probe volume filtering is more visible and important! So unless you know how much filtering you have due to the probe volume averaging and how this affect the spectra, you cannot just compare the slopes of the measured lidar spectra with classical turbulence slopes like the –5/3 one.

8. Section 3.3 also does not make sense to me (title should not be stress but variance by the way). Unless I just missed the values of pitch and roll different from zero, the results in Fig. 5 should be perfect (zero bias and R = 1 when plotting the vertical variance and beam 5 variance as Eq. (10) shows). So why are they not perfect? It just seems very strange to me.

**Minor comments**

1. Line 2 and other instances in the paper: you can remove "(light detection and ranging)". Lidars as radars and sodars are already "workds" and do not need explanation

2. Line 5 Replace "compared to that derived" to "compare to those derived"

3. Line 27 and other instances in the paper: replace "altitude" by "height"

4. Line 32 Replace "This system" by "lidars"

5. Line 40 the explanation of the cross-contamination is not true. This is due to the influence of the different velocity components, which result from the way a lidar normally scans

6. Line 42 I think you do not mean TKE spectra, but velocity component spectra

7. Line 59 Add "the" before instrument

8. Line 79 Add "the" before Doppler

9. Line 82 and maybe other instances "40 meters to 300 meters" should read "40 to 300 m"

10. Line 94 sometimes you say "beam" and sometimes "Beam", be consistent and check all instances

11. Equation (6) appears alone in the text

12. Equations (8)–(10) should appear after the "are given" in line 32

13. Line 139 Two "involves" should be "involved"

14. Line 145 delete the s in measurements in this and other instances

15. Line 146 Flat terrain does not mean you will get good atmospheric conditions. Maybe you mean orography undisturbed flow or similar

16. Figure 3 y-axis is not TKE but beam power spectra density or similar

17. I am not sure the right word is "distributions" for what you show in Fig. 6. In line 277 you say TI distributions are not governed by wind speed. Well they should if you did plot the TI as function of wind speed

18. Line 323 not true (see main comment 1)

19. Line 335 vertical resolution and probe length are not the same!

20. Line 378 and around. A large probe lenght increases largely the filtering, which is perhaps the biggest threat of lidars

**References**

R. Dewey and S. Stringer. Reynolds stresses and turbulent kinetic energy estimates from various ADCP beam configurations: Theory. *J Phys Ocean*, pages 1–35, 2007.

W. L. Eberhard, R. E. Cupp, and K. R. Healy. Doppler lidar measurement of profiles of turbulence and momentum flux. *J. Atmos. Ocean. Tech.*, 6:809–819, 1989.

W. Fu, A. Peña, and J. Mann. Turbulence statistics from three different nacelle lidars. *Wind Energ. Sci*, 7:831–848, 2022.

A. Peña, J. Mann, and N. Dimitrov. Turbulence characterization from a forward-looking nacelle lidar. *Wind Energ. Sci*, 2:133–152, 2017.

A. Sathe, J. Mann, N. Vasiljevic, and G. Lea. A six-beam method to measure turbulence statistics using ground-based wind lidars. *Atmos. Meas. Tech. Discuss.*, 7:10327–10359, 2014.

---

## Author Comment (AC1)

**Deriving atmospheric turbulence intensity from profiling pulsed lidar measurements - wes-2022-53**

**Response on reviewer's comments - RC1**

**October 21, 2022**

Thank you to the three reviewers (RC1/RC2 and CC1) for their very valuable comments on our manuscript. We rewrote more than half of the paper to satisfy their recommendations. The main changes appear in blue in the revised version. The main changes are the following:

- In the introduction, a paragraph has been added to put into context the noise (doppler noise) as it is defined in the present paper and the noise (signal-to-noise) addressed in the literature.

- In "Data collection and methods", a full section is now dedicated to the definition of the Doppler noise. The influence of the cell size and sampling rate of the magnitude of the Doppler noise are addressed. A step by step procedure is proposed to evaluate the Doppler noise and its variance that induces overestimation of TI. In addition, the cutting frequency, set to 80% of the Nyquist frequency in the first version of the manuscript is now determined by a method involving an error minimization of the least-square regression of the LOS velocity spectra.

- Section 3.1 (turbulent kinetic energy spectra) and section 3.3 (Vertical stress) have been removed. Section 3.1 was considered out of the scope of this paper by the reviewers. Section 3.3 was considered irrelevant.

- Mathematical expressions of $\overline{u'^2}$ and $\overline{v'^2}$ do not consider the pitch and roll anymore. The deployment of both lidars were done such as the pitch and roll were almost zero, there are thus negligible. Without pitch and roll, expressions of $\overline{u'^2}$ and $\overline{v'^2}$ match the classic expressions proposed by [1]. $\overline{w'^2}$ was removed because it has been considered irrelevant for this study.

- A stationarity study of the 10-min subsets of the LOS velocity time series is proposed through the Augmented Dickey-Fuller test.

- The discussion and conclusion have been completely rewritten. Limits of the variance method are discussed. The "benefice" of the higher sampling rate is also discussed with regards to other limitations of lidars such as the probe-volume averaging. Recommendations are made to improve the next generation of lidars such as the addition of extra beams and simultaneous acquisition of the LOS velocities.

**1  General comments**

The manuscript "Deriving atmospheric turbulence intensity from profiling pulsed lidar measurements" by Thiébaut et al. deals with the estimation of the turbulence intensity of the horizontal wind velocity component using the 5-beam DBS scanning mode of the WindCube v2.1 lidar. The manuscript focuses first on the turbulent kinetic energy (TKE) spectrum and then assesses the accuracy of the different methods and lidar configurations to study the turbulence intensity using various metrics.

The manuscript addresses challenging, contemporary and interesting questions about turbulence measurements at heights relevant to large wind turbines. Therefore, the paper is of broad international interest and within the scope of Wind Energy Science (WES), although it may be better suited to Atmospheric Measurement Techniques (AMT). The paper brings a novel idea for the calculation of the velocity variance, inspired by ocean science, which is welcome. The objective and methods are clearly outlined.

However, the analysis and method need some improvements as they sometimes rely on unverified or incorrect assumptions. Some preliminary steps lack rigour. More critically, the discussion and results are sometimes self-contradictory, for example, the discussion on the probe length. The content can be more concise. For example, section 3.1 (turbulent kinetic energy spectra) diverges sometimes from the initial scope of the paper. Section 3.3 could be summarized in a couple of sentences. Even if the paper is smoothly written and read quite easily, a major revision is, therefore, needed. In summary, the manuscript relies on a good idea, but the method, results and discussion need considerable improvements to be considered for publication in WES.

**2  Specific comments**

**Point 1**

The turbulence intensity (TI) can be calculated for the three wind velocity components: $I_u$, $I_v$ and $I_w$ with $I_u > I_v > I_w$. For wind turbine design, the along-wind component $u$ and cross-wind components $v$ are the most important. In the manuscript, eq (3) gives the turbulence intensity of the horizontal components, i.e. $u^2 + v^2$, which is of limited interest for wind turbine design. Therefore, I suggest focusing on the $u$ and $v$ components separately.

**Reply**

We agree that the along and cross-wind components are the most important for wind turbine design. Our first idea was to rotate the variance $\overline{u'^2}$ and $\overline{v'^2}$ such as the velocity component $u_{\text{rot}}$ is aligned with the mean wind direction and $v_{\text{rot}}$ is forced to 0. The rotated variance are described as follows [2, 3]:

$$\overline{u'^2}_{\text{rot}} = \overline{u'^2}\sin^2\Theta + \overline{v'^2}\cos^2\Theta + \overline{u'v'}\sin 2\Theta \tag{1}$$

$$\overline{v'^2}_{\text{rot}} = \overline{u'^2}\cos^2\Theta + \overline{v'^2}\sin^2\Theta - \overline{u'v'}\sin 2\Theta \tag{2}$$

where $\Theta$ is the mean wind direction and the subscript "rot" refers to variance components in the rotated coordinate system. Equations (1) and (2) involve the shear stress $\overline{u'v'}$. This is the only component of the Reynolds stress tensor that remains unknown from WindCube 2.1 measurement. However, $\overline{u'v'}$ can be calculated from measurement of the sonic anemometer. For example, in [3], the authors derived $\overline{u'v'}$ from sonic anemometer measurements and found that $\overline{u'v'}$ was near zero throughout the day and thus had a negligible effect on the coordinate rotation, thus justifying to neglect the term $\overline{u'v'}$ in Equations (1) and (2). In our study, however, we decided to use the sonic anemometer measurements only for validation of TI and not for validation of assumptions since the trend in the offshore wind community is to use lidars (and lidar only) instead of anemometers mounted on tower to perform the site assessment stage. We added a sentence specifying this point (Line 165). We also mentioned in the discussion (updated version of the manuscript) that a possible way to apply the variance method on the along and cross wind velocity time series will be the addition of a sixth beam to the lidar as it has been done in Ocean science with ADCP. This will allow to resolve the six components of the Reynolds stress tensor.

**Point 2**

The manuscript correctly mentions that the TI is a central and practical metric for the study of atmospheric turbulence. However, the velocity spectrum is ultimately needed for the wind loading calculation. Therefore, the importance of the TI for wind energy and wind engineering should not be overestimated. This aspect may be pointed out in the introduction of the manuscript.

**Reply**

We tempered the impact of TI for wind energy applications in the introduction with these new sentences: "The key parameters are the turbulence intensity, TI, i.e., the ratio of velocity fluctuations to the velocity mean, and the turbulent kinetic energy (TKE) spectra, i.e., the velocity variance as a function of frequency. Both parameters are directly involved in modeling applications of wake effects within wind farms which can significantly decrease the power production and increase the cost of electricity (Howland et al., 2019)". Lines 17-20.

**Point 3**

Lines 29-30: I suggest not to write that lidars have emerged as an alternative to met masts but rather as a "complementary tool". This is particularly true for turbulence measurements: both profilers and scanning Doppler wind lidars are currently unable to match the performances and data availability of 3D sonic anemometers.

**Reply**

We changed "alternative" by "complementary". Line 30.

**Point 4**

Line 41: The manuscript argues that the along-beam spatial averaging effect will not be addressed. I think it should, on the contrary, be studied. Several of the questions raised by the manuscript can be linked to the fact that the along-beam spatial averaging is not discussed jointly with the sampling frequency, for example.

**Reply**

This point is now addressed in the discussion (Lines 305-321) and conclusion (Lines 361-366). As you mentioned in Point 5, it is likely that the increase of the frequency rate by a factor 4 in comparison to the commercial lidar, is still not enough to bring any improvement in TI estimates. The eddies, and their associated variance, that are supposed to be captured with the higher sampling rate might remains undetectable because of the probe-volume averaging.

**Point 5**

In the manuscript, arguments for using a higher sampling frequency are given. I agree that increasing the sampling frequency $f_s$ should helps reducing the statistical uncertainties (see e.g. [4] (Section 7.1, 1994)). However, because of the along-beam spatial averaging of 20 m or more, increasing $f_s$ to values higher than 1 Hz (or even 0.25 Hz?) may not help to capture smaller eddies.

**Reply**

We replied to this remark in Point 4. After reading the section you recommended, we agree that the "better" performance of the prototype configuration in TI reconstruction is probably due to the reduce statistical uncertainties that brings the higher sampling rate. We now mentioned this point in the discussion. Lines 322 - 326.

**Point 6**

Section 2.2: A sketch to show the different wind velocity components in the wind-based coordinate system ($u$, $v$ and $w$) and the lidar-based coordinate system $V_x$, $V_y$, $V_z$ would be a good idea for the sake of pedagogy.

**Reply**

We added a sketch in Figure 2 (left hand-side figure) showing the orientation of the beam 1, 2, 3 and 4. Beam 1 is aligned with $u$ and $V_x$ and beam 4 is aligned with $v$ and $V_y$.

**Point 7**

Section 2.3.2: It is a little unclear to me where the pitch and roll angle comes from. I understand that these angles are based on the publication by [5] about the ADCP system, but I don't know if these are necessary for a Doppler wind lidar. If yes, how were they measured? were they non-negligible?

**Reply**

Both pitch and roll are given by the lidar interface during the deployment and there are also available in the header of each .rtd files. A proper deployment requires to set the pitch and roll to values close to zero. In our deployment, the pitch and roll of the commercial configuration were set to -0.126˚ and 0.021˚ respectively and the pitch and roll of the prototype configuration were set to -0.74˚ and 0.23˚ respectively. We agree that in the case of bottom-fixed lidar installation the pitch and roll are negligible. We wanted to stick to the full expressions proposed by Dewey and Stringer. In the revised version of the manuscript, the expressions are simplified by considering that the pitch and roll are negligible (Eq. 12 and 13).

**Point 8**

As far as I know, the study by Dewey and Stringer [5] is actually unpublished. The reference used in the manuscript seems to be given by Google Scholar, which may be incorrect.

**Reply**

You're correct. The study by Dewey and Stringer has never been published. However, this manuscript is more and more employed in the field of ocean science since the development of a new generation of ADCP employing five beams instead of four. The fifth beam being oriented vertically upward.

**Point 9**

Eq. 10 seems to be unnecessary complex since one could argue that $w \approx b_5$. Section 3.3. of the manuscript seems to show that assuming $w \approx b_5$ is good enough. Therefore, I suggest moving and squeezing the content of section 3.3. right after equation 10. In general, one wants to avoid presenting an excessively heavy formula for the sake of clarity.

**Reply**

We removed this equation and the section 3.3 which are unnecessary according to the two others reviewers.

**Point 10**

Section 2.4.1: The company Thies Clima produces different types of sonic anemometers. It is unclear to me which model was used. Note that if a 2D sonic anemometer was used, one should keep in mind that this type of sensor is mainly used as a weather station. So this sensor is not expected to perform as well as a 3D sonic anemometer to study turbulence. This is an important point to remember if the goal is to validate Doppler wind lidar turbulence measurements against sonic anemometry.

**Reply**

The sonic anemometer is a 3D Thies (No. 4.3830). We added this information on the updated version of the manuscript. Line 207.

**Point 11**

Fig 2 contains elements that are too small to be easily readable. I suggest redrawing it. Ideally, the figure should indicate whether the sonic anemometer is on the northwest or southeast side of the tower. From the text, it seems it is on the southeast side. Maybe a digital Elevation Map can be used instead of satellite images? Otherwise, it can be difficult to visualize the local topography.

**Reply**

We redraw the Fig. 2 following your suggestion. There is now a map showing the local topography.

**Point 12**

Line 170: The comparison between sonic anemometer data and the lidar data is done using the same sampling frequency $f_s$ (0.25 Hz and 1 Hz). I agree with the authors on this approach. However, it also means that the turbulence intensity will be underestimated compared to a standard 3D sonic anemometer using $f_s > 10$ Hz. This should be clarified in the manuscript.

**Reply**

The downsampling will indeed bring underestimation of TI. We illustrated this point by calculating TI derived from time series sampled at 4 Hz and downsampled at 1 Hz and 0.25 Hz respectively. This gave the respective values of 7.4%, 7.2% and 6.95%. Lines 228-231.

**Point 13**

Lines 172-178: There is no guarantee that a 10 min duration is short enough to ensure stationarity of the velocity fluctuations. To assess the assumption of stationarity, a trend test (non-parametric) or a parametric test should be used.

**Reply**

We added a section dedicated to the stationarity study of the 10-min subsets of the LOS velocity time series (section 2.7). The stationarity has been evaluated by the Augmented Dickey-Fuller (ADF) test. ADF tests the null hypothesis that a unit root is present in a time series sample. The interpretation of the result is done using the p-value given from the test. A p-value of less than 5% means the test rejects the null hypothesis, thus, the time series is stationary. p-value of the 10-min LOS velocity time series associated each beam of the commercial and prototype configurations was found varying within the range [1.1%, 2.3%], thus, we can argue that the 10-min LOS velocity time series are stationary and that the 10-min temporal window is of sufficient length to perform turbulence analysis of the present wind dataset.

**Point 14**

Lines 172-178: The standard deviation $\sigma_H$ of the horizontal wind velocity component $H$ is studied using an averaging time of 10 min. For turbulence measurements, averaging times of 30 min to 60 min are often used. Considering the Simiu and Scanlan spectrum for the along-wind component $u$, it can be shown that using 10 min instead of 60 min (30 min) will lead to an underestimation of $\sigma_u$ by 15% (11%).

The standards and codes for wind turbine design use often 10 min (i.e. 600 s) as averaging time because the wind loading is computed utilizing the velocity spectra. Considering frequencies down to

$\frac{1}{600s} = 0.00167$ Hz is often good enough to describe the full range of vibrations of a large engineering structure. However, it does not mean that 10 min is long enough to study integral turbulence characteristics.

**Reply**

We totally agree on this point. We performed our analysis on 10-min subsets only because the temperature and pressure at our disposal were 10-min averaged values. We needed these values to calculate the potential temperature and classify the dataset according to stable and unstable atmospheric conditions. Otherwise, we would have chosen to perform the turbulence analysis on 30-min subsets.

**Point 15**

Line 186: Does the 75% data availability means that time series with 25% or less of NaNs were kept? if yes, this may be too much. I suggest dismissing samples if the percentage of Nans is 10% or higher. In atmospheric science, the acceptable percentage of NaNs is usually from 2.5% to 5%, but this may be too strict for Doppler wind lidar data. As a result, 326 more subsets were removed from the analysis.

**Reply**

Yes, 75% data availability means that time series with 25% or less of NaNs were kept. However, we followed your recommendation and set the threshold of data rejection to 90% of data availability within each 10-min subset.

**Point 16**

Lines 187-188: I do not understand the sentence "A percentage ranging [...] was rejected". May it be possible to clarify it?

**Reply**

We rephrased this: "For the commercial configuration, the maximum percentage of subsets rejection, i.e., 1.7%, was associated with the beam 4 whereas the minimum percentage, i.e., 0.6%, was associated with the beam 1. For the prototype lidar, these percentages were found to be more than twice higher". Lines 207-209.

**Point 17**

Section 2.6: For the study of the gradient Richardson number, I recommend using the potential temperature $\theta$ or virtual potential temperature $\theta_v$ instead of the absolute temperature $T$. From figure 2 in the manuscript, the pressure and humidity seem to be measured at 10 m and 95 m, so $\theta$ and $\theta_v$ may be calculated successfully. It may also be a good idea to use a more accurate classification of atmospheric turbulence than $R_i < 0$ or $R_i > 0$, which could be too rough.

**Reply**

Your comment is in line with a comment of the second reviewer. In the revised version of the manuscript, the classification was done through the study of the sign of the vertical gradient of the potential temperature, $d\theta/dz$. A convective unstable wind flow is associated with $d\theta/dz < 0$ while stable wind flow is associated with $d\theta/dz > 0$ as proposed in [6]. Our first idea was to decompose the dataset into 3 atmospheric classes (unstable, stable and strongly stable) instead of only 2. However, just few subsets ($\approx 50$) were recorded during strongly stable conditions. It is a too low number in comparison to the number of subsets recorded during stable and unstable conditions. Thus, the comparison of TI would have been biased.

**Point 18**

Section 3.1: This section contains some unnecessary sentences (lines 211-215) and may have to be re-written to better anchor it to the research question. Figure 3 should be redrawn. It is vital to better highlight the presence (or absence) of an inertial subrange. Because of the low-sampling frequency, the cross-contamination and the spatial averaging, the inertial subrange may not be easily visible.

To improve the visualization of the velocity spectra: (1) the velocity spectra should be split between $u$ and $v$ components, and (2) the velocity spectra should be pre-multiplied with the frequency or wavenumber. If needed, you can further normalize the spectra using the variance $\sigma_u^2$ measured by the sonic anemometer at 95 m; (3) The frequency should be replaced by the wavenumber or a normalized frequency (4) the velocity spectra should be split between unstable or stable stratifications.

**Reply**

The section 3.1, and thus the spectra, have been removed in the updated version of the manuscript. However, the mean spectrum of the LOS velocities associated with the beam 1 of the commercial and prototype lidars is shown. The mean spectra are shown to illustrate the step-by-step method for the Doppler noise identification. For this step, an identification of the cutting frequency is required, thus justifying the representation of the frequency instead of the wavenumber. The representation of spectra recorded during stable and unstable conditions is now irrelevant.

**Point 19**

Lines 236-243: These lines seem unnecessary. They could be removed without affecting the content of the paper.

**Reply**

These lines have been removed.

**Point 20**

Have you tried estimating the standard deviation of the u and v components by integrating a fitted empirical velocity spectrum as

$$\sigma_u^2 = \int_0^\infty S_u(f) df \tag{3}$$

where $f$ is the frequency in Hz and $S_u$ has the following form:

$$f S_u(f) = \frac{A f_r}{(1 + B f_r)^{5/3}} \tag{4}$$

$$f_r = \frac{fz}{\overline{u}} \tag{5}$$

where $A$ and $B$ are empirically obtained by least-square fit to the estimated spectrum and $u$ is the mean wind speed at height $z$. If yes, how does this method compares with the noise-removal approach adopted in the present study? Note that you may have to apply a method similar to [7] to reduce the cross-contamination in the velocity spectrum.

Following [7], the velocity spectra from DBS scans should not be fitted by turbulence models. Although they do not specify what they call "turbulence models", one can assume they refer to the uniform shear model or 3D isotropic spectral turbulence models. I ignore if their recommendation for "turbulence models" includes eq. (4). Nevertheless, attempting to estimate $\sigma_u$ using eq. (3) and eq. (4) may be worthwhile, especially when the lidar system is aligned with the mean wind direction.

**Reply**

We did not try to solve the standard deviation by fitted an empirical velocity spectrum. However we plan to test this approach with the measurements we are going to collect in Le Planier Island, in the Mediterranean Sea. The lidar will be deployed such that one beam, let's say beam 1, will be aligned

with the Mistral to get direct measurements of the turbulence structures associated with this specific wind event. With the removal of the Doppler noise, the eq. (3) would become (for beam 1):

$$\sigma_1^2 = \int_0^\infty [S_1(f) - N_1] df \tag{6}$$

where $N_1$ is the constant spectral density associated with the Doppler noise. $N_1$ would be determined following section 2.3 of the revised version of the manuscript. The result would the fitting of a spectrum whose energy is slightly lower than the spectrum fitted with eq. 3. However, and you mentioned this point, it is not recommended to fit a spectrum derived from DBS measurements, specifically the part where probe-volume averaging is the most important, i.e., the inertial domain and its associated -5/3 slope. Unless we know how much filtering we have due to the probe-volume averaging and how this affect the spectra, we cannot just compare the slopes of the measured lidar spectra with classical turbulence slopes like the –5/3 one. See major comment n°7 of the reviewer CC1.

**Point 21**

Fig 4: It is unclear to me what the figure aims to demonstrate. If the quadratic relationship is not used elsewhere in the manuscript, should it be kept? Is the $R^2$ value shown in this figure the Pearson or Spearman correlation coefficient? I suggest using the Spearman correlation coefficient if the relationship between the two variables is non-linear monotonic. Alternatively, the RMSE can be used instead.

**Reply**

The fitting of the scatterplots is irrelevant. We removed it. The purpose of this figure is to show that the Doppler noise increases with increasing wind speed and that the prototype configuration gives Doppler noise lower than the commercial configuration probably because the noise is distributed along a wider frequency range.

**Point 22**

Fig 6 does not seem to clearly support the conclusions of the manuscript. Was it because there was no noise removal here? I encourage the use of a colormap that is perceptually uniform instead of the jet colormap.

**Reply**

Maybe it wasn't clear in the conclusion. We rewrote this part. Globally the variance method gives half of TI estimates that are overestimated because of the cross-contamination effect which means that during certain wind conditions the main source of error is the cross-contamination, whereas, the other half is underestimated which means that the volume averaging is the mean source of error. In figure 6 (first version), now figure 7, includes a removal of the Doppler noise. This generates scatter TI estimates because the removal of the Doppler noise is not made on individual fluctuations but rather on the mean fluctuations averaged over each 10-min subset. Thus, the noise correction is sensitive to the number of realizations considered and Doppler noise will always result in some spreading of the corrected TI estimates. That is the main drawback of this method. Moreover, we redraw the figure and used a uniform color map instead of the jet colormap.

**Point 23**

Section 4: The discussion seems to recommend a large probe length to study turbulence. In general, when turbulence is studied with a Doppler wind lidar, one wants the probe volume length to be as small as possible. A 20 m probe volume length is already quite large, which is the reason continuous-wave Doppler (scanning) lidars with a probe volume length smaller than 10 m have a higher potential than pulsed lidar to study turbulence. Therefore, this section may need some reformulations.

**Reply**

Our recommendation was awkward. We agree. It is relevant for acoustic Doppler current profiler (ADCP) deployment in ocean science since the cell size are configurable and can be set to only few dozen of centimeters. This recommendation is not appropriate for the 20 m probe length of the WindCube. A recommendation would rather be the reduction of the probe length to capture smaller eddies. This will induce a higher level of noise whose variance can be relatively well evaluated and remove from turbulence estimates.

**Point 24**

Lines 381-394: This paragraph seems more adapted to the beginning of the manuscript since it reviews some previous results. Maybe this can be moved there and shortened?

**Reply**

The discussion has been completely rewritten. We removed this paragraph.

**Point 25**

The conclusion may have to be reformulated as it includes several recommendations that could be criticized. I agree that operating a lidar at a higher sampling rate is a good idea, but this will not be useful if the probe volume is not reduced. As pointed out by the authors, reducing the probe volume increases, in return, the measurement noise. So the situation is rather complex. Does it mean that DBS scans should only be used to study the mean flow characteristics? Should new scanning modes be developed instead?

Also, the conclusion mentions the use of lower beam inclination $\phi$ to improve the measurement accuracy. This may be a good idea. However, if $\phi$ becomes too small, the measurement uncertainty will increase substantially because the angle between the beam and the horizontal direction will get close to 90°. So the ideal value of the beam inclination $\phi$ is not trivial either. Do you have any specific value in mind?

**Reply**

The conclusion has been completely rewritten. We now agree that a faster sampling rate would have a limited interest as soon as the probe length remains the same. A reduction of the probe length would be the priority before eventually considering an increase of the sampling rate.

DBS scans is indeed relevant to study mean flow characteristics but it comes with several limitations to study the wind fluctuations. Instead of swinging from one beam to another (DBS technique), a new technique would be to collect LOS velocity simultaneously as it is done with ADCP in ocean science. This has the potential to limit or annihilate the cross-contamination effect which generate systematic overestimation of TI due to the two-point correlation of different wind field components. Addition of extra beams should also be investigated. This would enable resolving the full Reynolds stress tensor thus allowing the application of the variance method to provide TI estimates associated with the along and cross-wind components as required by the wind energy industry.

Moreover, as you said, the reduction of the beam inclination, $\phi$ is not trivial. The angle of 28° is optimum in terms of accuracy of the measured wind speed and direction and it is unlikely that effort would be put on reducing this angle for the next generation of WindCube. Note that the beam inclination of ADCP, for example the Nortek Signature, is 25°.

**Point 26**

The manuscript contains some typographical errors. I recommend a quick proofread. The online web app Grammarly (https://app.grammarly.com) is quite good for this purpose.

**Reply**

We did a quick proofread. Thank you for proposing the web app Grammarly. Very useful.

**References**

[1] W. L. Eberhard, R. E. Cupp, and K. R. Healy, "Doppler lidar measurement of profiles of turbulence and momentum flux," *Journal of Atmospheric and Oceanic Technology*, vol. 6, no. 5, pp. 809–819, 1989.

[2] A. Sathe, J. Mann, N. Vasiljevic, and G. Lea, "A six-beam method to measure turbulence statistics using ground-based wind lidars," *Atmospheric Measurement Techniques*, vol. 8, no. 2, pp. 729–740, 2015. Publisher: Copernicus GmbH.

[3] J. F. Newman, P. M. Klein, S. Wharton, A. Sathe, T. A. Bonin, P. B. Chilson, and A. Muschinski, "Evaluation of three lidar scanning strategies for turbulence measurements," *Atmospheric Measurement Techniques*, vol. 9, no. 5, pp. 1993–2013, 2016. Publisher: Copernicus GmbH.

[4] J. C. Kaimal and J. J. Finnigan, *Atmospheric boundary layer flows: their structure and measurement*. Oxford university press, 1994.

[5] R. Dewey and S. Stringer, "Reynolds stresses and turbulent kinetic energy estimates from various ADCP beam configurations: Theory," *J. of Phys. Ocean*, pp. 1–35, 2007.

[6] D. L. Hartmann, *Global physical climatology*, vol. 103. Newnes, 2015.

[7] F. Kelberlau and J. Mann, "Cross-contamination effect on turbulence spectra from Doppler beam swinging wind lidar," *Wind Energy Science*, vol. 5, no. 2, pp. 519–541, 2020. Publisher: Copernicus GmbH.

---

## Author Comment (AC2)

**Deriving atmospheric turbulence intensity from profiling pulsed lidar measurements - wes-2022-53**

**Response on reviewer's comments - RC2**

**October 21, 2022**

Thank you to the three reviewers (RC1/RC2 and CC1) for their very valuable comments on our manuscript. We rewrote more than half of the paper to satisfy their recommendations. The main changes appear in blue in the revised version. The main changes are the following:

- In the introduction, a paragraph has been added to put into context the noise (doppler noise) as it is defined in the present paper and the noise (signal-to-noise) addressed in the literature.

- In "Data collection and methods", a full section is now dedicated to the definition of the Doppler noise. The influence of the cell size and sampling rate of the magnitude of the Doppler noise are addressed. A step by step procedure is proposed to evaluate the Doppler noise and its variance that induces overestimation of TI. In addition, the cutting frequency, set to 80% of the Nyquist frequency in the first version of the manuscript is now determined by a method involving an error minimization of the least-square regression of the LOS velocity spectra.

- Section 3.1 (turbulent kinetic energy spectra) and section 3.3 (Vertical stress) have been removed. Section 3.1 was considered out of the scope of this paper by the reviewers. Section 3.3 was considered irrelevant.

- Mathematical expressions of $\overline{u'^2}$ and $\overline{v'^2}$ do not consider the pitch and roll anymore. The deployment of both lidars were done such as the pitch and roll were almost zero, there are thus negligible. Without pitch and roll, expressions of $\overline{u'^2}$ and $\overline{v'^2}$ match the classic expressions proposed by [1]. $\overline{w'^2}$ was removed because it has been considered irrelevant for this study.

- A stationarity study of the 10-min subsets of the LOS velocity time series is proposed through the Augmented Dickey-Fuller test.

- The discussion and conclusion have been completely rewritten. Limits of the variance method are discussed. The "benefice" of the higher sampling rate is also discussed with regards to other limitations of lidars such as the probe-volume averaging. Recommendations are made to improve the next generation of lidars such as the addition of extra beams and simultaneous acquisition of the LOS velocities.

**1 General comments**

Thiébaut et al. describe a new method to estimate turbulence intensity from profiling lidar measurements. Such studies are quite relevant for site assessment in wind energy. Better estimates of turbulence intensity are very important for load predictions. It is well known that profiling lidars have some shortcomings when it comes to turbulence estimation and new methods to improve the results are thus welcome. The idea in this manuscript to use methods from acoustic Doppler current profilers (ADCP) which are used in oceanic sciences is interesting and attractive. Three main changes are made to commercial profiling lidars: the sampling rate is increased, the variance method as used with ADCPs is implemented and a noise removal is applied. While the increase of sampling rate is straight forward and the explanations are easy to follow, the critical differences between ADCP and profiling lidars for atmospheric measurements are not sufficiently well elaborated. For this reason, I cannot suggest the paper for submission in WES before some major revisions.

**Point 1**

The main difference between ADCP and lidar DBS is the character of the atmosphere, which can be much more instationary than ocean currents, and the fact that a lidar DBS does not measure all beams simultaneously. A minor difference is that the onshore lidar as it is used in this study does not move, so it has a constant roll and pitch angle which equals zero in best case. It is unclear why the authors use the full roll and pitch equation from Dewey and Stringer (2007) and not the much more simple zero pitch and roll equations. From those equations, it would also be very easy to check if the assumptions of homogeneity and stationarity are valid for the dataset by checking the equations 11 and 12 from Dewey and Stringer against the sonic anemometer data:

$$\overline{b_1'^2} + \overline{b_2'^2} = 2\overline{u'^2}\sin^2\phi + 2\overline{w'^2}\cos^2\phi \tag{1}$$

$$\overline{b_3'^2} + \overline{b_4'^2} = 2\overline{v'^2}\sin^2\phi + 2\overline{w'^2}\cos^2\phi \tag{2}$$

Does the left hand side as measured from the lidar DBS equal the right hand side as measured from the sonic anemometer for different wind speeds and different stability conditions? Only checking the vertical wind is not sufficient.

**Reply**

Mathematical expressions of $\overline{u'^2}$ and $\overline{v'^2}$ do not consider the pitch and roll anymore. The deployments of both lidars were done such as the pitch and roll were almost zero, there are thus negligible. Without pitch and roll, expressions of $\overline{u'^2}$ and $\overline{v'^2}$ match the classic expressions proposed by [1]. $\overline{w'^2}$ was removed because it has been considered as irrelevant for this study.

Your proposition of using eq. (7) and (8) above is indeed a good idea. However, the vertical velocity were not recorded by the sonic anemometer to avoid overload of the data logger. Thus, we cannot check the homogeneity and stationarity following your suggestion. In the revised version of the manuscript, a stationarity study of the 10-min subsets of the LOS velocity time series is proposed through the Augmented Dickey-Fuller test.

**Point 2**

The definition of Doppler noise seems to be a quite vague to me. Parts of the explanation are given in different parts of the manuscript, but it should be better introduced in the beginning. It is not true that lidar noise has not been studied and included in models for turbulence retrieval from lidars before. The work by R.G. Frehlich as well as I. Smalikho and E.J. O'Connor contain much information about lidar noise. The noise which is meant here should be put into context to these other studies.

**Reply**

In the revised version of the manuscript, a full section (section 2.3) is now dedicated to the definition of the Doppler noise. The influence of the cell size and sampling rate of the magnitude of the Doppler

noise are mentioned. A step by step procedure is proposed to evaluate the Doppler noise and its variance that induces overestimation of TI.

In the introduction, the noise as it is defined in the present manuscript is put into context. Previous works have been investigated noise in term of signal-to-noise (SNR). Lines 48 - 54.

**Point 3**

The way that stability is derived from the Richardson number here is not correct for atmospheric sciences. What is calculated is the bulk Richardson number, which has a non-zero critical value between stable and unstable flow. This has an impact on the whole comparison between stable and unstable conditions in this study and needs to be revised.

**Reply**

Your comment is in line with a comment of the first reviewer. In the revised version of the manuscript the classification is done through the study of the sign of the vertical gradient of the potential temperature, $d\theta/dz$. A convective unstable wind flow is associated with $d\theta/dz < 0$ while stable wind flow is associated with $d\theta/dz > 0$ as proposed in [2]. The decomposition proposed in the first version of the manuscript was wrong. Now, 72.7% of the 10-min subsets are associated with unstable conditions whereas 27.3% are associated with stable conditions.

**2 Specific comments**

- p.2,l.40: what about availability?

  **Reply**

  Thank you, we added this limitation. Line 42.

- p.2, ll.51f: lidar noise is explored quite extensively in works by R.G. Frehlich, I. Smalikho and E.J. O'Connor. These works should be reflected.

  **Reply**

  This comment is addressed in Point 2.

- p.4,l.83: "volume of atmosphere"? This is a strange expression.

  **Reply**

  This expression has been removed.

- Fig.2: basic information of the maps are missing: scale, northing, lat/lon, copyright of the map pictures.

  **Reply**

  Fig. 2 has been modified following your suggestion.

- p.8,l.170: I do not understand the reason for the resampling of the sonic. Is it not the goal to compare the lidar derived TI to the best possible measurement? If the sonic is downsampled, it will also lack very small scale turbulence. Probably the difference is not much, but I would suggest to use the best possible TI estimate for comparison, unless you want to isolate the errors by specific processing steps.

  **Reply**

  We wanted to isolate the error due to the different sampling rates. The first reviewer agree on this approach (see point 12 - RC1). We added three sentences in the revised version: "Due to the resample, the reference TI estimates presented in this paper are underestimated. To illustrate this point, the median of the reference TI was calculated from the 10-min subsets sampled at 4 Hz and downsampled at 1 Hz and 0.25 Hz. This gave the respective values of 7.4%, 7.2% and 6.95%". Lines 195-198.

- p.8,l.174: it is a questionable statement that 10-minutes are enough to retain the longest ime scales of coherent turbulent structures. In basic ABL research, 30-minutes are rather the standard.

**Reply**

This comment is in line with a comment of the first reviewer (Point 13 - RC1). We agree that 30-min is the standard. We performed our analysis on 10-min subsets only because the temperature and pressure at our disposal were 10-min averaged values. We needed these values to calculate the potential temperature and classify the dataset according to stable and unstable atmospheric conditions. Otherwise, we would have chosen to perform the turbulence analysis on 30-min subsets.

We added a section dedicated to the stationarity study of the 10-min subsets of the LOS velocity time series (section 2.7). The stationarity has been evaluated by the Augmented Dickey-Fuller (ADF) test. ADF tests the null hypothesis that a unit root is present in a time series sample. The interpretation of the result is done using the p-value given from the test. A p-value of less than 5rejects the null hypothesis, thus, the time series is stationary. p-value of the 10-min LOS velocity time series associated each beam of the commercial and prototype configurations was found varying within the range [1.1%, 2.3%], thus, we can argue that the 10-min LOS velocity time series are stationary and that the 10-min temporal window is of sufficient length to perform turbulence analysis of the present wind dataset.

- p.8, l.197: It is too much simplified to use the measured temperature difference and standard atmosphere lapse rate for the calculation of the Richardson number. Please calculate the potential temperature difference.

**Reply**

In the revised version, we calculated the potential temperature to classify our dataset. See response to your comment "Point 3".

- p.9, l.205: It might be partly because of the way the Richardson number was calculated, but it is certainly a strong overestimation if 89.4% of the data are considered to be measured in a stable atmosphere. What is actually calculated here is the bulk Richardson number which has a critical value of 0.25 for unstable flows.

**Reply**

Exactly, it was a strong overestimation. We made a mistake here which has been fixed in the revised version.

- p.11, ll.242f: I am less concerned about the inertial subrange being present at frequencies higher than the Nyquist frequency, but more concerned if the integral length scale is large enough to yield an inertial subrange at the resolved frequencies, especially in stable atmospheric conditions. Typically, for lidar turbulence retrievals, there is a minimum integral length scale below which the estimates need to be discarded.

**Reply**

Absolutely. With a pulsed lidar employing DBS scan, we think that the limited turbulence length scale that can be measure accurately in the light-of-sight direction is the probe length (20 m for the WindCube). However, for turbine design specifications, turbulence length scales associated with the wind horizontal wind components are the most relevant. The characterization of these scales and their associated turbulent energy requires the combination of beams. Thus, when characterizing horizontal turbulence, the beam spread is the limiting length scale. Due to the diverging beams, the volume, proportional to the beam spread $\Delta b$, in which the measurements are being integrated increases with increasing altitude, changing the volume averaging of the turbulence metrics. A recommendation would be to restrict the analysis to length scales (and corresponding frequencies) that are higher than the beam spread. The beam spread is a function of height, $H$, above ground such that $\Delta b = 2H\tan\phi$. Considering the beams inclination, $\phi =$

28°, of the WindCube v2.1, the beam spread between two opposite beams at 97-m altitude is 103 m which matches the land-based wind turbines rotor diameter. Since wind turbines respond to turbulence on scales similar to the rotor diameters [$\mathcal{O}(100)$ m] one can say that the commercial configuration of the WindCube v2.1 is able to measure accurately the large horizontal turbulent structures of interest for wind energy applications.

- p.11, l.247: does $N_i$ and $n_i$ have to be determined for each beam individually? How much do they differ and why?

    **Reply**

    Yes, $N_i$ and $n_i$ needs to determined for each beam individually. In ocean science, it has been shown with ADCP measurements that the Doppler noise increases with increasing flow speed [3, 4]. For lidar measurement, it is the same. We demonstrated this (Fig. 5 in the revised version). For the commercial configuration, the maximum percentage of subsets rejection, i.e., 1.7%, was associated with the beam 4 whereas the minimum percentage, i.e., 0.6%, was associated with the beam 1. For the prototype lidar, these percentages were found to be more than twice higher (Lines 207 - 209).

    It is difficult to tell why the the Doppler noise is not constant and instead, depend on the flow speed. The Doppler noise is generated by random scatterer motions within the sample volumes which results to errors in measuring the frequency change or phase shift of the reflected pulses. We would say that at higher flow speed the scatter motions are more "chaotic" which might impact the error in measuring the frequency change.

- p.11, l.248: this seems to be a random choice for the characteristic frequency. Can this be justified by error quantification?

    **Reply**

    In the revised version of the manuscript (section 2.3), the cutting frequency is now determined by a method involving an error minimization of the least-square regression of the LOS velocity spectra.

- p.11, l.252: I assume you mean Eq. 5.

    **Reply**

    Yes. It has been corrected.

- p.17, l.363: what does "a frequency domain 9 times wider" mean?

    **Reply**

    We removed this sentence.

- p.18, ll.370ff: These explanations and theories would be much easier to follow and understand if the Doppler noise was presented in a concise mathematical formulation before.

    **Reply**

    We added a section dedicated to Doppler noise (section 2.3) to help the reader following the explanations.

- p.19, l. 411: Reducing the beam spread needs to be carefully traded off against horizontal wind speed retrieval accuracy.

    **Reply**

    You are right. The angle of 28° is optimum in terms of accuracy of the measured wind speed and direction and it is unlikely that effort would be put on reducing this angle for the next generation of WindCube. Lines 319 - 321.

**References**

[1] W. L. Eberhard, R. E. Cupp, and K. R. Healy, "Doppler lidar measurement of profiles of turbulence and momentum flux," *Journal of Atmospheric and Oceanic Technology*, vol. 6, no. 5, pp. 809–819, 1989.

[2] D. L. Hartmann, *Global physical climatology*, vol. 103. Newnes, 2015.

[3] J. Thomson, B. Polagye, V. Durgesh, and M. C. Richmond, "Measurements of turbulence at two tidal energy sites in Puget Sound, WA," *Oceanic Engineering, IEEE Journal of Oceanic Engineering*, vol. 37, no. 3, pp. 363–374, 2012.

[4] M. Thiébaut, J.-F. Filipot, C. Maisondieu, G. Damblans, R. Duarte, E. Droniou, N. Chaplain, and S. Guillou, "A comprehensive assessment of turbulence at a tidal-stream energy site influenced by wind-generated ocean waves," *Energy*, vol. 191, p. 116550, 2020.

---

## Author Comment (AC3)

**Deriving atmospheric turbulence intensity from profiling pulsed lidar measurements**

**Response on reviewer's comments - RC3**

**October 21, 2022**

Thank you to the three reviewers (RC1/RC2/RC3) for their very valuable comments on our manuscript. We rewrote more than half of the paper to satisfy their recommendations. The main changes appear in blue in the revised version. The main changes are the following:

- In the introduction, a paragraph has been added to put into context the noise (doppler noise) as it is defined in the present paper and the noise (signal-to-noise) addressed in the literature.

- In "Data collection and methods", a full section is now dedicated to the definition of the Doppler noise. The influence of the cell size and sampling rate of the magnitude of the Doppler noise are addressed. A step by step procedure is proposed to evaluate the Doppler noise and its variance that induces overestimation of TI. In addition, the cutting frequency, set to 80% of the Nyquist frequency in the first version of the manuscript is now determined by a method involving an error minimization of the least-square regression of the LOS velocity spectra.

- Section 3.1 (turbulent kinetic energy spectra) and section 3.3 (Vertical stress) have been removed. Section 3.1 was considered out of the scope of this paper by the reviewers. Section 3.3 was considered irrelevant.

- Mathematical expressions of $\overline{u'^2}$ and $\overline{v'^2}$ do not consider the pitch and roll anymore. The deployment of both lidars were done such as the pitch and roll were almost zero, there are thus negligible. Without pitch and roll, expressions of $\overline{u'^2}$ and $\overline{v'^2}$ match the classic expressions proposed by [1]. $\overline{w'^2}$ was removed because it has been considered irrelevant for this study.

- A stationarity study of the 10-min subsets of the LOS velocity time series is proposed through the Augmented Dickey-Fuller test.

- The discussion and conclusion have been completely rewritten. Limits of the variance method are discussed. The "benefice" of the higher sampling rate is also discussed with regards to other limitations of lidars such as the probe-volume averaging. Recommendations are made to improve the next generation of lidars such as the addition of extra beams and simultaneous acquisition of the LOS velocities.

**1 Major comments**

Thanks a lot for your manuscript. I have several mayor comments as well as minor comments that I think are mostly related to misconceptions with regards to the nature of measuring turbulence with lidars and lack of clarity.

**Point 1**

In the abstract, as an example, the authors say that a new method is proposed to estimate TI from pulsed profilers. First, the method the authors used was proposed by Dewey and Stringer [2]. Second, the use of beam variances have also been used before to estimate velocity-component variances at the least by both profilers and nacelle-based lidars [3, 4]. So the novelty should be clarified.

**Reply**
The novelty has been clarified. In the revised version, the abstract starts with "A method developed in ocean science and based on acoustic Doppler current profiler (ADCP) measurements is implemented to provide estimates of the atmospheric turbulence intensity (TI) derived from measurements of pulsed lidars employing the Doppler beam swinging technique". Lines 1-3.

**Point 2**

It could be the writing style but I am not sure whether the authors understand the problem of measuring turbulence with a lidar. For example, the cross-contamination is not due to the different structures of the field (line 40) but due to the influence of different velocity components on the line-of-sight variance, which is a result of the lidars scanning strategy. Also, in line 45 they say that cross-contamination causes and overestimation of TI. That is not always true; it might happen but that depends on how much filtering due to probe volume averaging you have. So, in many cases, if not most cases, filtering is the biggest threat in lidars.

**Reply**
We rewrote this part of the paper by mentioning that there are two systematic errors when measuring turbulence with a lidar: "In comparison to TI derived from measurement of a reference instrument such as a sonic anemometer, TI derived from lidar measurements is biased by two main systematic errors, i.e., underestimation due to the probe-volume averaging, and overestimation due to the cross-contamination effect causes by the influence of different velocity components on the line-of-sight (LOS) variance, which is a result of the lidars scanning strategy". Lines 38-41. Also, in the discussion we mentioned that both sources of error do not cancel each other.

**Point 3**

Also important is that by acquiring the velocity faster the change in the variance should not be high and so neither in the TI. You will increase the uncertainty on the variance by measuring slower but the bias should not change unless you definitively and systematically are missing fluctuations by the turbulence structures but this is not the case of this unit measuring this non-complex flow.

**Reply**
You're right. The increase of the sampling rate does not bring any significant improvement in TI estimates. Although the increase of the sampling rate has brought TI estimates slightly closer to that given by the reference measurements, this might be the result of a reduced statistical uncertainty generated by a higher number of measurement points within the 10-min subsets (Kaimal and Finnigan, 1994) rather than the ability of the prototype configuration to capture the variance associated with smaller eddies. These eddies might still be filtered out due to the probe-volume averaging.

**Point 4**

Instrument noise corrections have been explored before (line 53) (e.g., [5]). Perhaps you could explore that method as in that work they study a pulsed lidar too. This would avoid using a threshold to

establish the frequency at which you expect noise, which is what I think you are doing. Your method does seem very sensitive to this choice and you should show how sensitive or not indeed is.

**Reply**
In the revised version of the manuscript (section 2.3), the cutting frequency is now determined by a method involving an error minimization of the least-square regression of the LOS velocity spectra.

**Point 5**

You have defined TI to be the parameter you want to look for. However, you should also present mean and variance comparisons of the velocity components as the problems with turbulence should be clearer seen when computing the variance and you want to make sure both lidars measure the same mean wind as your sonic. Also in wind energy the TI is normally defined based on the along-wind component or horizontal wind. If you have your fixed lidar beams then the horizontal velocity variance $\sigma_S^2$ is not $(\sigma_x^2 + \sigma_y^2)/2$ as you imply but $(V_x^2\sigma_x^2 + V_y^2\sigma_y^2)/S^2$ where $S$ is the horizontal velocity magnitude in the case the covariance between the two horizontal components is assumed zero (which is probably ok in your case). So I wonder why you choose to define TI like this. Perhaps you can make a comparison of sonic variance as you imply (half of the sum of both variances) against deriving the variance from the horizontal velocity time series from the sonic.

**Reply**
The definition of the two-dimensional (2D) TI was found in the literature (e.g., [6, 7]) where the 2D TI is given by:

$$\mathrm{TI}_{2D} = 100 \times \sqrt{\frac{\frac{1}{2}\left(\sigma_x^2 + \sigma_y^2\right)}{\overline{V_x}^2 + \overline{V_y}^2}} \tag{1}$$

A comparison of $\sigma_S^2 = (\sigma_x^2 + \sigma_y^2)/2$ and $\sigma_S^2 = (V_x^2\sigma_x^2 + V_y^2\sigma_y^2)/S^2$ is shown in Fig. 1. Both time series are almost superimposed.

**Point 6**

My most important comment is the use of the method by Dewey and Stringer [2]. I cannot find what values of pitch and roll the authors use in Eqs. (8)–(10). If they use close to zero values (as I guess

[Figure]

Figure 1: Comparison of $\sigma_S^2 = (\sigma_x^2 + \sigma_y^2)/2$ and $\sigma_S^2 = (V_x^2\sigma_x^2 + V_y^2\sigma_y^2)/S^2$.

they place the lidar relatively well), then those equations are not needed but the classical Eberhard expressions [1] for the beam velocity variance, which one can easily show lead to:

$$\langle u'u' \rangle = \left(\sigma_3^2 + \sigma_1^2 - 2\sigma_5 \cos^2\phi\right) / \left(2 \sin^2\phi\right) \tag{2}$$

$$\langle v'v' \rangle = \left(\sigma_2^2 + \sigma_4^2 - 2\sigma_5 \cos^2\phi\right) / \left(2 \sin^2\phi\right) \tag{3}$$

For $\langle w'w' \rangle$ you do not need an expression as you measure with a vertical beam (so you can see that when $\phi_1$ and $\phi_2$ are close to zero in Eq. (10), the variance of $w$ and that of beam 5 are the same).

**Reply**
Your comment is in line with comments of the two other reviewers. In our deployment, the pitch and roll of the commercial configuration were set to -0.126° and 0.021° respectively and the pitch and roll of the prototype configuration were set to -0.74° and 0.23° respectively. We agree that in the case of bottom-fixed lidar installation the pitch and roll are negligible. We wanted to stick to the full expressions proposed by Dewey and Stringer. In the revised version of the manuscript, the expressions are simplified by considering that the pitch and roll are negligible (Eq. 12 and 13).

**Point 7**

Section 3.1. makes little sense to me. First in Fig. 3 you seem to be plotting the beam power spectra and not the TKE spectra as you state in the text. Second, and most important, you are here analyzing the slope of the spectra within the frequency range where the probe volume filtering is more visible and important! So unless you know how much filtering you have due to the probe volume averaging and how this affect the spectra, you cannot just compare the slopes of the measured lidar spectra with classical turbulence slopes like the –5/3 one.

**Point 8**

Section 3.3 also does not make sense to me (title should not be stress but variance by the way). Unless I just missed the values of pitch and roll different from zero, the results in Fig. 5 should be perfect (zero bias and R = 1 when plotting the vertical variance and beam 5 variance as Eq. (10) shows). So why are they not perfect? It just seems very strange to me.

**Reply for both points 7 and 8**
Sections 3.1 and 3.3 have been removed. Yourself and the two others reviewers have considered these two sections as out of the scope of the paper and/or irrelevant.

**2  Minor comments**

- Line 2 and other instances in the paper: you can remove "(light detection and ranging)". Lidars as radars and sodars are already "works" and do not need explanation.

  **Reply**
  'light detection and ranging" has been removed.

- Line 5 Replace "compared to that derived" to "compare to those derived".

  **Reply**
  Done. Thank you.

- Line 27 and other instances in the paper: replace "altitude" by "height".

  **Reply**
  Done for each instance.

- Line 32 Replace "This system" by "lidars".

  **Reply**
  Done

- Line 40 the explanation of the cross-contamination is not true. This is due to the influence of the different velocity components, which result from the way a lidar normally scans.

  **Reply**
  Thank you. We corrected the definition: "In comparison to TI derived from measurement of a reference instrument such as a sonic anemometer, TI derived from lidar measurements is biased by two main systematic errors, i.e., underestimation due to the probe-volume averaging, and overestimation due to the cross-contamination effect causes by the influence of different velocity components on the line-of-sight (LOS) variance, which is a result of the lidars scanning strategy". Lines 38-41.

- Line 42 I think you do not mean TKE spectra, but velocity component spectra.

  **Reply**
  Exactly. It has been corrected.

- Line 59 Add "the" before instrument.

  **Reply**
  This part has been deleted.

- Line 79 Add "the" before Doppler.

  **Reply**
  This part has been deleted.

- Line 82 and maybe other instances "40 meters to 300 meters" should read "40 to 300 m".

  **Reply**
  This part has been deleted. Changed for other instances.

- Line 94 sometimes you say "beam" and sometimes "Beam", be consistent and check all instances.

  **Reply**
  We checked all instances and stick to "beam".

- Equation (6) appears alone in the text.

  **Reply**
  This equation is useless and has been removed.

- Equations (8)–(10) should appear after the "are given" in line 32.

  **Reply**
  This part has been deleted.

- Line 139 Two "involves" should be "involved".

  **Reply**
  Thank you. It has been corrected.

- Line 145 delete the s in measurements in this and other instances.

  **Reply**
  Done for each instance.

- Line 146 Flat terrain does not mean you will get good atmospheric conditions. Maybe you mean orography undisturbed flow or similar.

  **Reply**
  Thank you for the proposition, we modified the text following your suggestion.

- Figure 3 y-axis is not TKE but beam power spectra density or similar.

  **Reply**
  It was a mistake, corrected in Fig. 3 and Fig. 4 of the revised version of the manuscript.

- I am not sure the right word is "distributions" for what you show in Fig. 6. In line 277 you say TI distributions are not governed by wind speed. Well they should if you did plot the TI as function of wind speed.

  **Reply**
  "distributions" might not be appropriate. We changed it by "estimates". In line 277, our idea was poorly expressed. It is, indeed, well known that TI depends on wind speed...

- Line 323 not true (see main comment 1).

  **Reply**
  See the answer to comment 1 (Point 1)

- Line 335 vertical resolution and probe length are not the same!

  **Reply**
  It is not indeed. Corrected.

- Line 378 and around. A large probe lenght increases largely the filtering, which is perhaps the biggest threat of lidars.

  **Reply**
  After the present study it is still difficult to tell either the volume-averaging or cross-contamination is the biggest threat.

**References**

[1] W. L. Eberhard, R. E. Cupp, and K. R. Healy, "Doppler lidar measurement of profiles of turbulence and momentum flux," *Journal of Atmospheric and Oceanic Technology*, vol. 6, no. 5, pp. 809–819, 1989.

[2] R. Dewey and S. Stringer, "Reynolds stresses and turbulent kinetic energy estimates from various ADCP beam configurations: Theory," *J. of Phys. Ocean*, pp. 1–35, 2007.

[3] A. Sathe, J. Mann, N. Vasiljevic, and G. Lea, "A six-beam method to measure turbulence statistics using ground-based wind lidars," *Atmospheric Measurement Techniques*, vol. 8, no. 2, pp. 729–740, 2015. Publisher: Copernicus GmbH.

[4] W. Fu, A. Peña, and J. Mann, "Turbulence statistics from three different nacelle lidars," *Wind Energy Science Discussions*, pp. 1–29, 2021. Publisher: Copernicus GmbH.

[5] A. Peña, J. Mann, and N. Dimitrov, "Turbulence characterization from a forward-looking nacelle lidar," *Wind Energy Science*, vol. 2, no. 1, pp. 133–152, 2017. Publisher: Copernicus GmbH.

[6] P. Mycek, B. Gaurier, G. Germain, G. Pinon, and E. Rivoalen, "Experimental study of the turbulence intensity effects on marine current turbines behaviour. Part I: One single turbine," *Renewable Energy*, vol. 66, pp. 729–746, 2014.

[7] M. Allmark, R. Martinez, S. Ordonez-Sanchez, C. Lloyd, T. O'doherty, G. Germain, B. Gaurier, and C. Johnstone, "A phenomenological study of lab-scale tidal turbine loading under combined irregular wave and shear flow conditions," *Journal of Marine Science and Engineering*, vol. 9, no. 6, p. 593, 2021. Publisher: MDPI.